# Exercising *D. melanogaster* Modulates the Mitochondrial Proteome and Physiology. The Effect on Lifespan Depends upon Age and Sex

**DOI:** 10.3390/ijms222111606

**Published:** 2021-10-27

**Authors:** Brad Ebanks, Ying Wang, Gunjan Katyal, Chloe Sargent, Thomas L. Ingram, Antonia Bowman, Nicoleta Moisoi, Lisa Chakrabarti

**Affiliations:** 1School of Veterinary Medicine and Science, University of Nottingham, Nottingham NG7 2RD, UK; mbybdpo@exmail.nottingham.ac.uk (B.E.); Wangying03@tyut.edu.cn (Y.W.); gunjan.katyal@nottingham.ac.uk (G.K.); Thomas.Ingram@md.catapult.org.uk (T.L.I.); svyab6@exmail.nottingham.ac.uk (A.B.); 2College of Biomedical Engineering, Taiyuan University of Technology, Taiyuan 030024, China; 3School of Bioscience, University of Nottingham, Nottingham NG7 2RD, UK; stxcs17@exmail.nottingham.ac.uk; 4Leicester School of Pharmacy, Leicester Institute for Pharmaceutical Innovation, De Montfort University, The Gateway, Leicester LE1 9BH, UK; nicoleta.moisoi@dmu.ac.uk; 5MRC Versus Arthritis Centre for Musculoskeletal Ageing Research, Nottingham NG7 2RD, UK

**Keywords:** mitochondria, ageing, exercise, *Drosophila*, respirometry, lifespan, proteomics

## Abstract

Ageing is a major risk factor for many of the most prevalent diseases, including neurodegenerative diseases, cancer, and heart disease. As the global population continues to age, behavioural interventions that can promote healthy ageing will improve quality of life and relieve the socioeconomic burden that comes with an aged society. Exercise is recognised as an effective intervention against many diseases of ageing, but we do not know the stage in an individual’s lifetime at which exercise is most effective at promoting healthy ageing, and whether or not it has a direct effect on lifespan. We exercised *w^1118^ Drosophila melanogaster*, investigating the effects of sex and group size at different stages of their lifetime, and recorded their lifespan. Climbing scores at 30 days were measured to record differences in fitness in response to exercise. We also assessed the mitochondrial proteome of *w^1118^ Drosophila* that had been exercised for one week, alongside mitochondrial respiration measured using high-resolution respirometry, to determine changes in mitochondrial physiology in response to exercise. We found that age-targeted exercise interventions improved the lifespan of both male and female *Drosophila*, and grouped males exercised in late life had improved climbing scores when compared with those exercised throughout their entire lifespan. The proteins of the electron transport chain were significantly upregulated in expression after one week of exercise, and complex-II-linked respiration was significantly increased in exercised *Drosophila*. Taken together, our findings provide a basis to test specific proteins, and complex II of the respiratory chain, as important effectors of exercise-induced healthy ageing.

## 1. Introduction

The dramatic ageing of the global population is a well-documented phenomenon. The World Health Organisation estimates that there are currently over 900 million people over the age of 60, and by 2050 this is set to increase to 2 billion [1]. With this demographic transformation, there will be huge economic costs incurred due to the health and social care needs of this group, in particular with the increased occurrence of non-communicable diseases. Within the EU, it has been estimated that over-65s already account for over 40% of healthcare spending—a figure that will increase as this demographic continues to expand [2]. It is therefore paramount that we develop a comprehensive understanding of the biology of ageing, and of how we can increase the healthspan of individuals as lifespans continue to rise.

Lifespan can be defined as the length of time between the birth and death of an individual; however, the definition of healthspan is different. One common definition of healthspan is ‘healthspan is the period of life spent in good health, free from the chronic diseases and disabilities of ageing’; the potential pitfalls of this definition are clear [3]. In humans, females live longer average lifespans in populations with both shorter and longer life expectancies [4]. Furthermore, lifestyle considerations can also influence the outcomes of lifespan and healthspan, and these can include whether an individual lives a solitary or social lifestyle. When considering the sociality of animals and lifespan, those that are obligately social tend to have greater lifespans when they have more associates and stronger social bonds with those associates [5,6,7,8]; however, this benefit is not universal, and in particular there are differences for species with facultative sociality [9,10,11].

The biology of ageing itself is a complex picture, but has characteristic hallmarks: genomic instability, telomere attrition, epigenetic alterations, loss of proteostasis, deregulated nutrient sensing, mitochondrial dysfunction, cellular senescence, stem cell exhaustion, and altered intercellular communication [12]. Due to the central role that functional mitochondria play in metabolism and intracellular signaling [13,14,15,16,17], the regulation of apoptosis [18], and their well-documented dysfunction in a broad range of age-related pathologies [19,20,21], a fundamental understanding of their role in the ageing process will anchor developments in the field.

Exercise is recognised as being pivotal in the fight to keep individuals healthier for longer. While exercise has long been advocated as a broad-spectrum remedy against ill health, many of the molecular details that underpin this have remained elusive. However, as the weight of evidence for exercise as a protective strategy accumulates, our comprehension of how exercise extends lifespan is increasing [22]. These molecular changes are now known to include exercise-induced oxidative stress, in line with mitohormesis [23,24].

*Drosophila* are a viable model for both ageing and lifespan studies, as well as mitochondrial studies [25,26]. This is due to their well-understood genetics, the high proportion of homologous genes that they share with *H. sapiens*—including those implicated in disease—and their relative ease in husbandry. Examples of lifespan assays in *Drosophila* include their use to explore the effects of D-GADD45, Cu/ZnSOD, and MnSOD overexpression, as well as survival outcomes [27,28]. However, lifespan assays have more traditionally been used in human and mammalian studies. Landmark mitochondrial studies in *Drosophila* were those that presented mechanistic evidence of PINK1 and parkin dysfunction in Parkinson’s disease [29,30,31].

Proteomics is the study of the entire complement of proteins present within a biological sample [32,33]. In this instance, we focused on the proteome of the mitochondrion, facilitating a deep evaluation of the physiology of the mitochondrion in both health and disease.

While it is recognised that exercise is an effective means of delaying the onset of certain hallmarks of ageing, and in preventing the onset of diseases of ageing, there is less certainty around how exercise should be applied as an intervention, or which molecular mechanisms are modulated to influence the ageing process. In this study, we explored exercise interventions at different stages of life for *Drosophila*, and recorded how this affected their probability of survival, plotting Kaplan–Meier survival curves with log-rank test analysis. To assess sex differences and the effects of a grouped or solitary lifestyle, we exercised female and male flies that were housed singly, as well as in groups of females and males. Climbing assay scores at 30 days were also measured to assess age-associated fitness correlated with exercise. To understand the molecular changes that are associated with differences in survival outcomes, we assessed the mitochondrial proteome via 2D gel electrophoresis coupled with mass spectrometry (2DE-MS) and label-free mass spectrometry of *Drosophila* that had been exercised for one week. We measured mitochondrial fitness via high-resolution respirometry to specifically reveal mitochondrial responses to exercise.

## 2. Materials and Methods

### 2.1. Fly Husbandry

We used the *Drosophila melanogaster* strain *w^1118^*. Fly food (Quick Mix Medium, Blades Biological) was added to the vial, to a depth of 1 cm, and 3 mL of distilled water was added; it was left for one minute, and then a small sprinkle of yeast was added. The singly housed *Drosophila* from further trials were given the same amount of food as those kept in groups. *Drosophila* were transferred to new food once a week, and were kept in an incubator at 25 °C. Food was kept hydrated with 150 μL of distilled water every day. When food became dry or the flies laid eggs, they were moved to a new vial and transferred back following food rehydration. The light cycle varied depending on when lights were turned off/on in the laboratory, but was generally a 12-h cycle. The study was reviewed and approved by the University of Nottingham SVMS local area ethics committee (#3091 200203 10 February 2020).

### 2.2. Fly Separation

Flies in glass vials were cooled on ice for 5–10 min, placed under a microscope to determine sex, and then placed into vials accordingly. The groups consisted of 20 flies in each vial. These were labelled GF/GM for exercised group females/males. Ten individual flies of known sex were placed in separate vials, labelled SF/SM for single females/males. Two cohorts of flies were exercised in independent experiments, and the datasets for the two cohorts were then pooled for the statistical analysis of the lifespan assay.

### 2.3. Exercise Regime

The methodology for the exercise regime was modified from the protocol used by Tinkerhess et al. [34]. The exercise machine utilised a power tower strategy that taps the flies down every 15 s to induce negative geotaxis behaviour. The flies were exercised within a 25 °C incubator. The flies were transferred to empty vials during the exercise sessions. Flies were exercised at 10:30 a.m. for 10 min on Mondays, Wednesdays, and Fridays, with a two-day rest on Saturdays and Sundays.

For lifespan assay, flies were subjected to one of the following five exercise regimes: lifetime exercise (weeks 1–6 of life), early-life exercise (weeks 1–2 of life), middle-life exercise (weeks 3–4 of life), late-life exercise (weeks 5–6 of life), or no exercise.

For proteomic analysis, mixed-sex flies that were 1–4 days post-eclosion were placed in vials of 20 flies and exercised for one week before being euthanized by freezing at −80 °C, one hour after the final exercise period.

### 2.4. Mortality

Deaths were recorded throughout the experiment. Mortality was analysed using GraphPad Prism 8 by inputting data into Kaplan–Meier survival graphs. Log-rank tests (*p* < 0.05) were carried out between groups of interest.

### 2.5. Climbing Assay

A modified power tower protocol was used to perform a ‘RING assay’ [35]. Flies were moved into empty vials, and images were taken 4 s after the frame had been tapped down. A climbing index score was calculated by multiplying the number of flies per quadrant score and dividing by the number of flies in the vial. Movie Player Classic was used to create frames to analyse, edited in Microsoft Paint to add lines and distinguish the 4 quadrants. A mean was calculated (*n* = 6), and unpaired *t*-tests were performed between means of different groups 30 days into their lifespan, using GraphPad Prism.

### 2.6. Tissue Preparation for HRR

High-resolution respirometry (HRR) analysis was carried out on the sixth day after the start of the exercise or control treatment. To prepare the tissue, flies were cooled on ice. before five were randomly chosen and mechanically homogenised in 500 µL of MiR05 buffer (Oroboros Instruments; 0.5 mM EGTA, 3 mM MgCl_2_, 60 mM lactobionic acid, 20 mM taurine, 10 mM KH_2_PO_4_, 20 mM HEPES, 110 mM D-sucrose, 1 g/L BSA, pH 7.1). The homogenate was briefly spun, and only the supernatant was used in order to exclude non-cellular debris. The sample was kept on ice until HRR analysis.

### 2.7. High-Resolution Respirometry

HRR was carried out using the Oroboros Oxygraph-2k (Oroboros^®^ Instruments, Innsbruck, Austria). The electrodes were calibrated daily to ensure that oxygen consumption was consistent, and analysis was carried out at 20 °C. A total of 100 μL of the fly homogenate was added to each chamber before the following substrates were added: (1) 5 µL and 10 µL of the complex I substrates, pyruvate and malate, respectively (5 mM and 2 mM, respectively); (2) 20 µL of 10 mM succinate—a complex II substrate; (3) 1 µL titrations of 0.5 µM CCCP; and, finally, (4) 1 µL of 2.5 µM antimycin A—a complex III inhibitor.

### 2.8. Mitochondrial Preparation

Flies were placed in 250 µL of mitochondrial isolation buffer, and then subjected to 5 min of manual homogenisation with a 1.2–2.0 mL Eppendorf micropestle. Fractions were obtained using protocols described previously [36].

### 2.9. 2D Gel Electrophoresis (2D–PAGE)

*Drosophila* mitochondrial fractions and whole-fly homogenates were subjected to isoelectric focusing using the ZOOM IPG system and pH 3–10 (non-linear) ZOOM IPG strips following the manufacturer’s protocol (Life Technologies, Carlsbad, CA, USA). Gels were stained and imaged before analysis with SameSpots software (one-way ANOVA). Spots identified with SameSpots were excised from the gel for proteomic analysis.

### 2.10. Matrix-Assisted Laser Desorption Ionization Tandem Time-of-Flight Mass Spectrometry (MALDI–TOF/MS)

Samples identified with SameSpots were excised from the 2D gel, and were analysed at the Centre of Excellence in Mass Spectrometry at the University of York. Proteins were reduced and alkylated, followed by in-gel digestion with trypsin. MALDI–TOF/MS was used to analyse the samples. The generated tandem MS data were compared against the NCBI database using the MASCOT search program to identify the proteins. De novo sequence interpretation for individual peptides was inferred from peptide matches.

### 2.11. Liquid Chromatography Tandem Mass Spectrometry (LC–MS/MS)

Following separation with SDS-PAGE (NuPAGE™ 4 to 12%, Bis-Tris, 1.0 mm, Mini Protein Gel, 12-well) according to the manufacturer’s protocols, the gel-resolved mitochondrial fractions were analysed at the Centre of Excellence in Mass Spectrometry at the University of York. Proteins were reduced and alkylated, followed by in-gel digestion with trypsin. LC–MS/MS was used to analyse the samples. Resulting LC–MS/MS data were imported in PEAKS Studio X for peak picking, peptide identification, and precursor-intensity-based relative protein quantification. Extracted tandem MS data were searched against the combined *D. melanogaster* and *S. cerevisiae* subsets of the UniProt database. Protein identifications were filtered to achieve a <1% false discovery rate (FDR), and to require a minimum of two unique peptides per protein group.

For relative label-free quantification, extracted ion chromatograms for identified peptides were extracted and integrated for all samples. Protein abundances were normalised between samples on the basis of the total area of identified peptide ions. Significance was established using the PEAKSX interpretation of the significance of the B model. The multiple tests corrected FDR thresholds. A −log10 *p* value > 23.52 (1% FDR) was deemed significant using the model.

### 2.12. Bioinformatic Analyses

To state that a protein had significant changes in abundance between the exercised and non-exercised groups, we developed the following criteria: the protein must be quantified in all three biological replicates, and the relative abundance ratios (RAR (exercise/non-exercise)) must be either <0.67 or >1.5 and have a −log10 *p* value > 23.52. Proteins that were not quantified in more than one biological triplicate in only one of the treatment groups were also noted as being of interest.

Two representative heatmaps were developed: one for proteins with increased abundance (RAR > 1.5), and one for proteins with decreased abundance (RAR < 0.67) post-exercise. All three biological replicates for each group were included, with relative abundances depicted using a red-to-blue (high-to-low relative abundance) colour gradient. Proteins were further analysed using the STRING database v.11.0 to examine functional relationships between proteins significantly different in abundance between the exercised and non-exercised groups. The platform was used to create two protein–protein interaction (PPI) networks. To narrow down proteins of interest, those with no associated interactions in the network were hidden, and protein nodes were coloured based on their biological process, molecular function, and cellular component designation as per Gene Ontology (GO) and UniProt annotation.

## 3. Results

### 3.1. In Males, Late-Life Exercise Has the Most Beneficial Effect, and Exercise throughout Life Is Detrimental in Comparison

Male *Drosophila* exercised throughout their lifetime had worse survival outcomes when compared with groups of males exercised at any other age or not exercised at all.

Grouped male *Drosophila* that were exercised throughout their lifetime had a worse probability of survival than those that were exercised in early life, middle life, or late life, as well as those that were not exercised at all (Figure 1A–D). Grouped male *Drosophila* that were exercised in middle life had an improved probability of survival compared with those that were not exercised at all (Figure 1E). Only single male *Drosophila* that were exercised during late life had an improved survival probability compared with those exercised throughout their lives (Figure 1F), which was also observed in the grouped male flies.

In groups of female Drosophila, age-targeted exercise was more beneficial than lifetime exercise; late-life exercise extended lifespan in individually housed female individuals.

Grouped female *Drosophila* exercised in early, middle, and late life had an increased probability of survival compared with those that were exercised throughout their lives (Figure 2A–C). As observed in male *Drosophila*, the single females that were exercised in late life also had an improved probability of survival compared with those that were exercised throughout their lives (Figure 2D). Single females that were not exercised at all also had an increased probability of survival compared with flies that were exercised throughout their lifetimes (Figure 2E), as well as those exercised in early life (Figure 2F).

When the climbing scores at 30 days were assessed for the *Drosophila* lifespan assay data in Figure 1 and Figure 2, we found that there were significant differences in climbing score outcomes (Figure 3). Flies that were exercised in the middle of their lives, along with flies not exercised at all, had lower climbing scores than flies that been exercised throughout their lifetimes. However, flies exercised in late life had improved climbing scores compared with those exercised throughout their lifetimes, correlating with the improved lifespan assay outcomes.

### 3.2. High-Resolution Respirometry of Exercised Drosophila

The flux control ratio (FCR) is an internal normalization of a given respiratory rate; It takes the ratio of the measured respiratory rate and the maximal uncoupled electron transport (ET) capacity of the mitochondria. We found a significant difference in the mean FCR of exercised and non-exercised flies when succinate—a complex II substrate—was added after the addition of pyruvate and malate (0.32 and 0.13, respectively; student’s *t*-test *p*-value = 0.025). Furthermore, the mean ET capacity was significantly greater in non-exercised flies compared to the exercised flies (42.55 and 14.27, respectively; Student’s *t*-test *p*-value = 0.002) (Figure 4, Table 1).

The spare respiratory capacity (SRC) is calculated as a relative value by (ET capacity specific flux)/(Routine specific flux * 100), where the SRC for non-exercised and exercised flies is 0.39 and 0.26, respectively.

### 3.3. There Is Increased Expression of Many Proteins in Response to Exercise, including Those of the Electron Transport Chain

The 2DE-MS method was used as a scoping technique to identify proteins of interest based on changes in their expression in response to exercise, as well as considering the effects of group size and sex of the flies (Appendix A). Label-free mass spectrometry was then used to determine differences in the expression of mitochondrial proteins after *Drosophila* had been exercised for one week. A total of 515 proteins were identified and quantified in relation to the whole sample.

Of the 515 proteins identified, 424 had significant differences in abundance between the groups that were analysed (>23.52 −log10P, >1% FDR). A total of 337 proteins were quantified in all three replicates of the exercised and non-exercised fly groups. It should be noted that one biological replicate from the exercised *Drosophila* group had less total protein than the other samples, but normalisation procedures compensated for this, and there was no notable effect on the results. Of the 337 commonly identified proteins, 51 had increased expression (RAR > 1.5) in response to exercise, with some intra-replica variation observed (Figure 5).

Using UniProt and Gene Ontology (GO) annotation for cellular component assignment, we found that 27 of the 51 proteins with increased expression in response to exercise localise to the mitochondria. The other proteins were from either the cytoplasm or other membrane-bound organelles, with likely close association with mitochondria resulting in their presence in the mitochondrial isolates.

Of the 27 mitochondrial proteins identified by GO:CC enrichment analysis, 24 are associated with the electron transport chain: 14 from complex I, 3 from complex III, 6 from complex IV, and 1 from cytochrome c-2 (Table 2).

### 3.4. A Subset of Mitochondrial Proteins Decreased in Response to One Week of Exercise in D. melanogaster

Of the 337 commonly identified proteins from the mitochondrial fraction, across all replicates from the exercise and non-exercise fly groups, there were 36 proteins that had significantly decreased expression in response to exercise (RAR < 0.67) (Figure 6).

Gene Ontology biological process (GO:BP) enrichment analysis with the STRING database v.11.0. identified that, among the proteins with significantly decreased expression in response to exercise, there was enrichment of a diverse range of metabolic pathways (Table 3). These pathways included the ATP metabolic process, the TCA cycle, the pyruvate metabolic process, the cellular amino acid metabolic process, the carboxylic acid metabolic process, and chaperone-mediated protein folding.

## 4. Discussion

### 4.1. Lifetime Exercise Has Worse Outcomes Than Targeted Exercise Interventions in Male and Female Drosophila

A consistent theme in our findings was the poor lifespan outcomes for both male and female *Drosophila* that were exercised throughout their lifetimes, as opposed to targeted interventions (Figure 1A–C,F and Figure 2A–D), or in some instances no exercise at all (Figure 1D and Figure 2E). It could be suggested that lifetime exercise of the flies could produce injury or exhaustion, leading to early mortality. In the case of single female flies, where no exercise at all had better lifespan outcomes than lifetime exercise, this theory is worth consideration.

Mechanistically, it is well understood that acute exercise induces oxidative stress through the generation of reactive oxygen species (ROS), which according to the mitohormetic explanation induce healthy levels of activity in cellular antioxidant responses [37]. The field has now moved away from the idea of oxidative stress being only detrimental with regards to ageing, with a more nuanced argument around low levels of oxidative stress and ageing [38]. However, it is also recognised that excessive levels of oxidative stress can contribute to the ageing process, so it may be the case that over-exercise—in this instance, lifetime exercise—results in detrimental levels of oxidative stress being produced as a cellular response, ultimately contributing to the early mortality of these flies [39,40]. As with every drug, the dose makes the poison.

It is noteworthy that, at 30 days in the lifespan assay, there was a large difference in the probability of survival for grouped male and female flies that were exercised throughout their lifetimes, compared with those that were subjected to targeted exercise interventions (Figure 1A–C and Figure 2A–C). This suggests that excessive, lifetime exercise can increase the risk of early mortality in the flies. Then, from the 30-day mark onward, the probability of survival in these groups falls to similar levels, which indicates that many of the detrimental effects of consistent exercise take place early in the lifespans of the flies.

### 4.2. Late-Life Exercise Produces a Rapid Improvement in Climbing Assay Scores Compared with Life-Time Exercise for Grouped Male Drosophila

Middle-life exercise, late-life exercise, and no exercise all produced significantly different climbing assay scores compared with lifetime exercise for grouped male *Drosophila* (Figure 3). When the difference seen between the ‘late life vs. lifetime’ and ‘no exercise vs. lifetime’ graphs is considered, it is striking given that the ‘late-life exercise group’ had only been subject to three days of exercise to this point. One possible explanation for this large swing in climbing scores in response to just three days of exercise is that there is a rapid, adaptive response to exercise. It has previously been observed that male *Drosophila* have a greater adaptive response to exercise than females, in age-matched 5-day-old flies [41]. It could be the case that the male flies exercised later in life also exhibit a rapid adaptive response to a similar, short amount of exercise training, which is no longer obvious after longer term exercise.

### 4.3. Succinate-Linked Respiration Is Elevated in Exercised Flies

We found that daily exercise in *D. melanogaster* significantly increases oxygen flux when succinate is supplied, compared with non-exercised flies (Figure 4). Succinate—a complex II substrate—is less efficient in producing ATP than complex-I-associated substrates (pyruvate and malate), as complex II does not pump protons that contribute to the electrochemical gradient. However, when ATP demand is high—such as during exercise—succinate may be an important substrate to help increase the ETC efficiency, due to complex I substrates being more rapidly diminished.

Succinate has been previously shown to be a respiratory substrate utilised during stress [42,43]. In the bumblebee, *Bombus terrestris*, succinate oxidation has been shown to cause a twofold increase in flight muscle mitochondria after a one-hour flight [42]. It is well established that exercise can induce an acute stress response; this resonates with our findings and the association between exercise, increased succinate oxidation, and complex II activity.

### 4.4. Reduced Spare Respiratory Capacity in Exercised Drosophila May Promote Longevity

Exercised flies had a significantly lower maximum electron transport chain capacity (ET capacity) compared to non-exercised flies (*p*-value = 0.002). Furthermore, the spare respiratory capacity (SRC)—described as the mitochondrial capacity to produce ATP beyond routine respiration—was higher in the non-exercised flies [44]. This suggests that non-exercised flies use a lower percentage of their maximum respiratory capacity to maintain routine respiration. This is in contrast to the findings of previous studies, as exercise is acknowledged to improve mitochondrial function [45].

One explanation for the lower SRC in the exercised flies could be due to acute stress induced by exercise, as low SRC has been associated with poor adaptation to stress and an inability to meet ATP demands [44]. However, the ability to meet the energetic requirements of the cell by utilising oxidative phosphorylation to the fullest extent before resorting to anaerobic means could be considered advantageous later in life, in that ‘if you don’t use it, you lose it’. Measurements are needed of acute and longer term exercise cohorts, for purposes of comparison.

### 4.5. Proteins from the Electron Transport Chain Are Significantly Upregulated in Response to Exercise

Proteomic analysis of mitochondria from flies that had been exercised for one week, 1–4 days post-eclosion, showed higher quantities of mitochondrial electron transport chain proteins (Table 2 and Appendix A). This could be a means of maximising the efficiency of the aerobic respiration that initially takes place during exercise during the transition to anaerobic respiration [46]. Enzymatic activities of the electron transport chain are decreased during the ageing process as markers of oxidative stress increase [47]. This may be a simplistic view, since Tavallaie et al. reported that administration of a moderate inhibitor of complex IV promoted mitochondrial fitness in C57BL/6J mice, suggesting that this may be used to mitigate metabolic syndromes of ageing in humans [48]. It is possible that an upregulated electron transport chain corresponds to greater mitochondrial fitness, which would be beneficial in the context of ageing.

### 4.6. Multiple Metabolic Pathways Are Downregulated in Response to Exercise

Enrichment analysis of proteins that were downregulated in the mitochondrial fraction after exercise pointed to a broad range of metabolic pathways (Table 3 and Appendix A). While it is difficult to suggest a direct link between these pathways and their decreased activity in exercise, the variety of pathways identified could simply reflect the cellular conservation and resource redirection that takes place during exercise-induced stress. The enrichment of chaperone-mediated protein folding proteins—specifically heat shock protein cognate 4 isoform G, and 60 kDa heat shock protein homolog 2 mitochondrial—may reflect a reduced rate of protein synthesis, which would also support this hypothesis. The identification of heat shock proteins is interesting, as these are connected with failed proteostasis—one of the hallmarks of ageing [12,49].

## 5. Conclusions

We found that targeted exercise, as opposed to lifetime exercise, produces better survival outcomes in male and female *Drosophila*. Exercise has a rapid and significant effect on mitochondrial physiology, as seen through changes in the ETC of fruit flies. Through proteomic analysis, we found that components of the electron transport chain are upregulated in response to exercise, while a variety of other metabolic pathways show decreased expression. We suggest that exercise causes increased utilisation of mitochondrial pathways, thus leading to better healthspan.

## Figures and Tables

**Figure 1 ijms-22-11606-f001:**
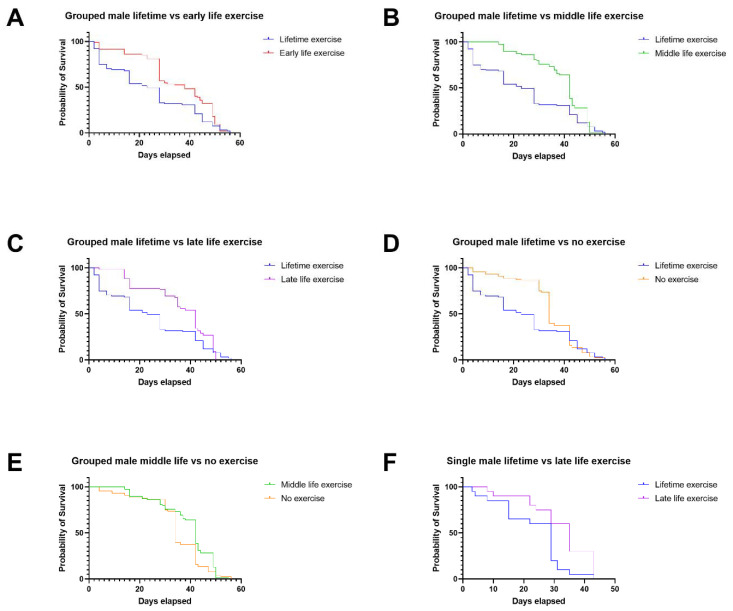
Percentage probability of survival for grouped and single male flies: (**A**) Grouped male flies’ lifetime exercise vs. early-life exercise (*n* = 93, 95, respectively) (log-rank test *p* = 0.0011). (**B**) Grouped male lifetime vs. middle-life exercise (*n* = 93, 78, respectively) (log-rank test *p* = 0.0014). (**C**) Grouped male lifetime vs. late-life exercise (*n* = 93, 121, respectively) (log-rank test *p* = 0.0050). (**D**) Grouped male lifetime vs. no exercise (*n* = 93, 87, respectively) (log-rank test *p* = 0.0237). (**E**) Grouped male middle-life vs. no exercise (*n* = 78, 87, respectively) (log-rank test *p* = 0.0125). (**F**) Single male lifetime vs. late-life exercise (*n* = 20, 20, respectively) (log-rank test *p* = 0.0053).

**Figure 2 ijms-22-11606-f002:**
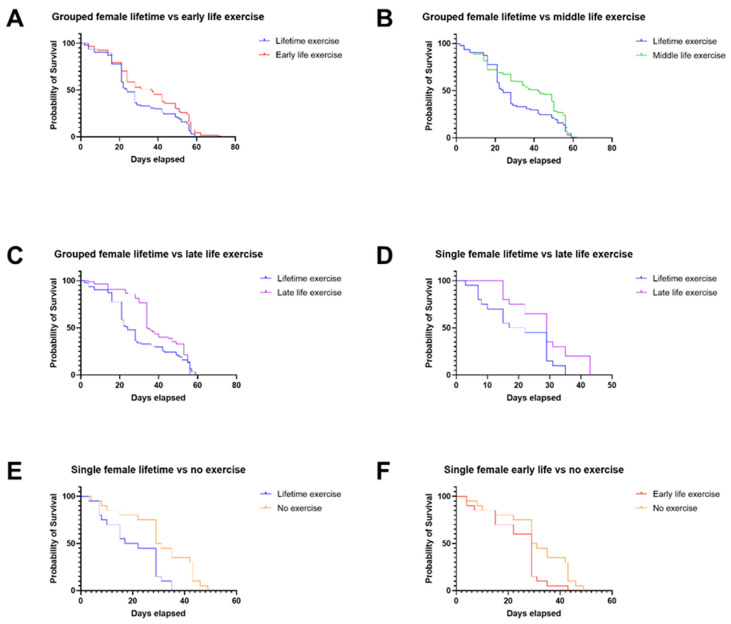
Percentage probability of survival for grouped and single female flies: (**A**) Grouped female lifetime vs. early-life exercise (*n* = 94, 121, respectively) (log-rank test *p* = 0.0094). (**B**) Grouped female lifetime vs. middle-life exercise (*n* = 94, 127, respectively) (log-rank test *p* = 0.0141). (**C**) Grouped female lifetime vs. late-life exercise (*n* = 94, 85, respectively) (log-rank test *p* = 0.0241). (**D**) Single female lifetime vs. late-life exercise (*n* = 20, 20, respectively) (log-rank test *p* = 0.0202). (**E**) Single female lifetime vs. no exercise (*n* = 20, 20, respectively) (log-rank test *p* = 0.0025). (**F**) Single female early-life vs. no exercise (*n* = 20, 20, respectively) (log-rank test *p* = 0.0202).

**Figure 3 ijms-22-11606-f003:**
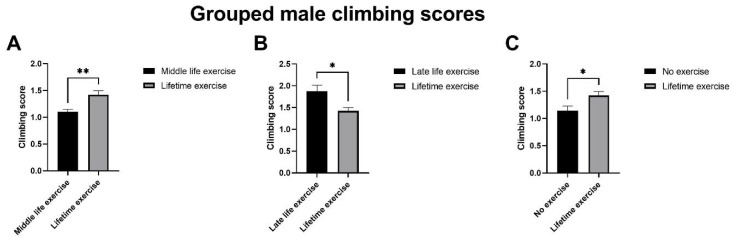
Grouped male Drosophila under different exercise regimens produced different climbing scores when compared with lifetime exercise: (**A**) Lifetime exercise vs. middle-life exercise (*n* = 6) (unpaired *t*-test, *p* = 0.0051, ** = <0.01). (**B**) Lifetime exercise vs. late-life exercise (*n* = 6) (unpaired *t*-test, *p* = 0.0183, * = < 0.05). (**C**) Lifetime exercise vs. no exercise (*n* = 6) (unpaired *t*-test, *p* = 0.0344, * = < 0.05).

**Figure 4 ijms-22-11606-f004:**
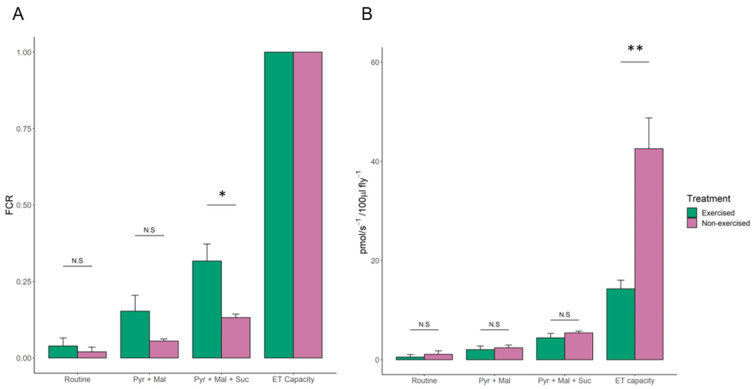
(**A**) Flux control ratio (FCR), in exercised and non-exercised male *D. melanogaster*; and (**B**) specific flux oxygen consumption in exercised and non-exercised male *D. melanogaster* (*n* = 3). Significant *p*-values: (**A**) succinate *p*-value = 0.025; (**B**) ET capacity *p*-value = 0.002. N.S = >0.05, * = <0.05, ** = < 0.01.

**Figure 5 ijms-22-11606-f005:**
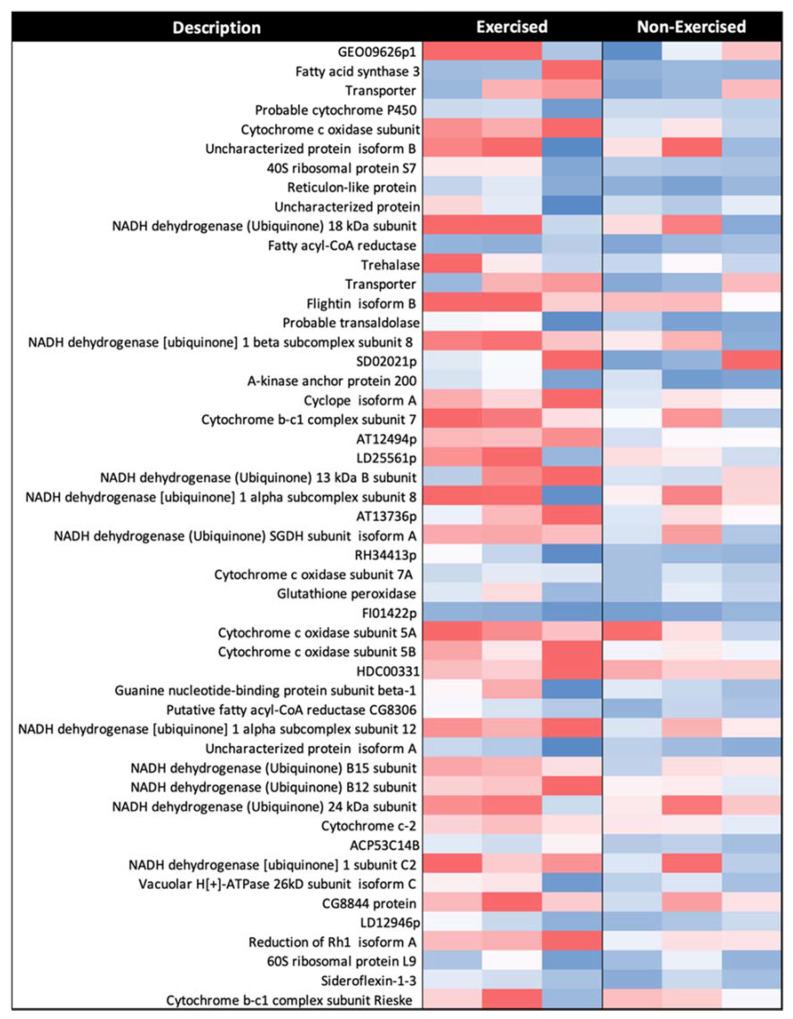
Representative heatmap of mitochondrial proteins with increased abundance (RAR of >1.5) after 1 week of exercise in *D. melanogaster* (*n* = 3). Red-to-blue colour gradient represents high-to-low relative protein abundance. The proteomic approach used was label-free mass spectrometry. All proteins listed were classified as significant as per the criteria described in the results.

**Figure 6 ijms-22-11606-f006:**
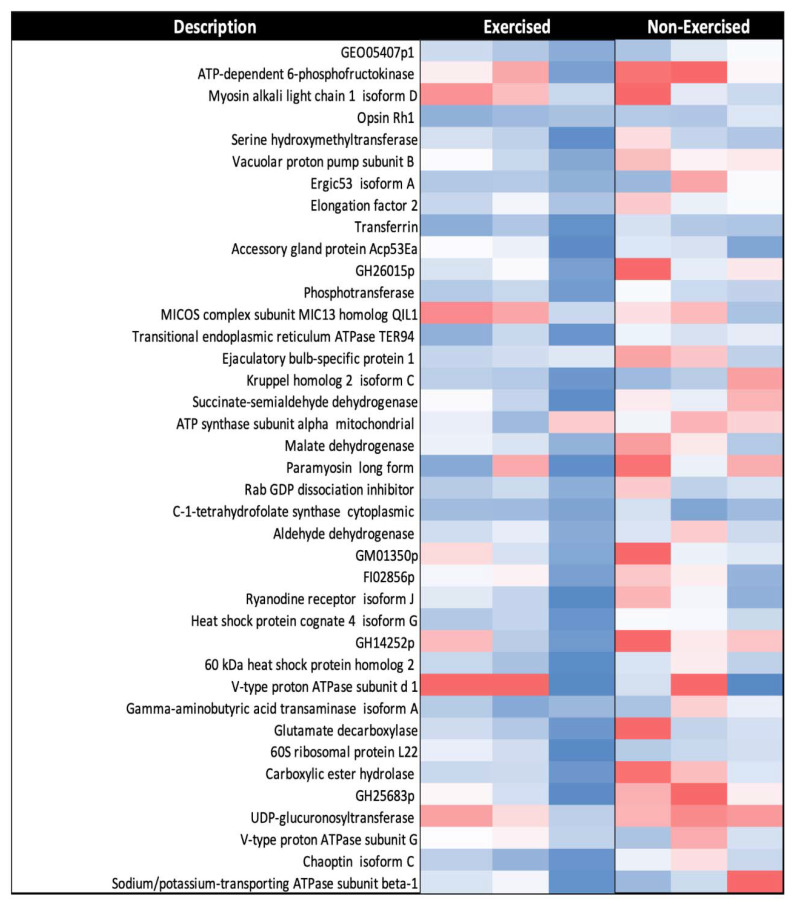
Representative heatmap of mitochondrial proteins with decreased abundance (RAR < 0.67) in response to one week of exercise in *D. melanogaster* (*n* = 3). Red-to-blue colour gradient represents high-to-low relative protein abundance. Proteomic analysis was conducted using label-free mass spectrometry. All proteins listed were classified as significant as per the criteria described in the results.

**Table 1 ijms-22-11606-t001:** Mean values of the specific flux and flux control ratio in exercised and non-exercised male *D. melanogaster* seen in Figure 4 (*n* = 3, * = *p* < 0.05; ** = *p* < 0.01). *p*-values from unpaired Student’s *t*-tests.

	Exercised	Non-Exercised	
	*n* = 5	*n* = 4	
Stage	Mean (s.e)	Mean (s.e)	*p*-Value
Routine			
Specific flux	0.55 (0.48)	1.08 (0.70)	0.537
FCR	0.04 (0.03)	0.02 (0.02)	0.58
Pyruvate & Malate			
Specific flux	2.04 (0.70)	2.41 (0.54)	0.699
FCR	0.15 (0.05)	0.05 (0.01)	0.143
Succinate			
Specific flux	4.40 (0.89)	5.39 (0.35)	0.381
FCR	0.32 (0.06)	0.13 (0.01)	0.025 *
ET capacity			
Specific flux	14.27 (1.77)	42.55 (6.19)	0.002 **

**Table 2 ijms-22-11606-t002:** Electron transport chain proteins increased in abundance in response to one week of exercise in *D. melanogaster* (*n* = 3). Enrichment analysis used Gene Ontology cellular component allocation from STRING database v.11.0. RAR: relative abundance ratio; FC: fold change.

Description		Accession	Significance (−log10P)	RAR	Log2 FC
Complex I					
NADH dehydrogenase (ubiquinone)	1Subunit C2	Q9VQM2	200	1.53	0.61
	1 Beta subcomplex subunit 8 mitochondrial	Q9W3X7	133.11	1.85	0.89
	1 Alpha subcomplex subunit 8	Q9W125	142.39	1.68	0.74
	1 Alpha subcomplex subunit 12	Q9VQD7	200	1.57	0.65
	SGDH subunit isoform A	Q9VTU2	86.17	1.64	0.71
	B15 subunit	Q6IDF5	36.57	1.56	0.64
	B12 subunit isoform A	Q9W2E8	88.95	1.55	0.64
	24 kDa subunit isoform A	Q9VX36	65.53	1.54	0.63
	18 kDa subunit	Q9VWI0	200	2.05	1.04
	13 kDa Bsubunit	Q9VTB4	78.35	1.69	0.76
GEO09626p1		Q8SYJ2	26.03	2.92	1.55
EG:152A3.7 protein		O97418	29.75	1.9	0.93
CG8844 protein		Q9VQR2	200	1.53	0.61
AT12494p		Q9VJZ4	70.67	1.7	0.77
Complex III					
Cytochrome b-c1 complex	Subunit Rieske mitochondrial	Q9VQ29	80.35	1.5	0.59
	Subunit 7	Q9VXI6	200	1.73	0.79
AT13736p		Q9VVH5	129.49	1.65	0.72
Complex IV					
Cyclope isoform A		Q9VMS1	200	1.78	0.83
Cytochrome C oxidase	Cytochrome C oxidase subunit	Q8IQW2	200	2.14	1.1
	Subunit 7A mitochondrial	Q9VHS2	82.59	1.61	0.69
	Subunit 5A mitochondrial	Q94514	110.61	1.6	0.68
	Subunit 5B isoform A	Q9VMB9	140.51	1.59	0.67
GEO09626p1		Q8SYJ2	26.03	2.92	1.55
Other OXPHOS proteins					
HDC00331		Q6IHY5	200	1.58	0.66
Uncharacterized protein isoform A		Q0KHZ6	93.18	1.56	0.64
Cytochrome c-2		P84029	38.14	1.54	0.62

**Table 3 ijms-22-11606-t003:** Proteins identified from the mitochondrial fraction with decreased expression in response to exercise, with Gene Ontology biological process enrichment, as determined by STRING database v.11.0. analysis. Enrichment of a diverse range of metabolic processes was identified among the proteins with decreased expression. RAR: relative abundance ratio; FC: fold change.

Description	Accession	Significance (−log10P)	RAR	Log2 FC
ATP metabolic process				
ATP synthase subunit alpha mitochondrial	P35381	84.91	0.62	−0.68
TCA cycle				
GM01350p	Q9VGQ1	32.66	0.58	−0.78
Malate dehydrogenase	Q9VKX2	56.18	0.62	−0.68
Pyruvate metabolic process				
Aldehyde dehydrogenase	Q9VLC5	38.53	0.59	−0.76
Cellular amino acid metabolic process				
Succinate-semialdehyde dehydrogenase	Q9VBP6	77.81	0.63	−0.67
Gamma-aminobutyric acid transaminase isoform A	Q9VW68	200	0.52	−0.94
Serine hydroxymethyltransferase	Q9W457	200	0.66	−0.59
Carboxylic acid metabolic process				
GM01350p	Q9VGQ1	32.66	0.58	−0.78
Aldehyde dehydrogenase	Q9VLC5	38.53	0.59	−0.76
Malate dehydrogenase	Q9VKX2	56.18	0.62	−0.68
Serine hydroxymethyltransferase	Q9W457	200	0.66	−0.59
Succinate-semialdehyde dehydrogenase	Q9VBP6	77.81	0.63	−0.67
Gamma-aminobutyric acid transaminase isoform A	Q9VW68	200	0.52	−0.94
Chaperone-mediated protein folding				
Heat shock protein cognate 4 isoform G	C7LA75	153.53	0.58	−0.8
60 kDa heat shock protein homolog 2 mitochondrial	Q9VMN5	200	0.55	−0.87
Other mitochondrial proteins				
MICOS complex subunit MIC60	A0A0B4KGN2	27.65	0.64	−0.65

## Data Availability

Please contact corresponding author.

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
