# Peer review of "Exercising D. melanogaster Modulates the Mitochondrial Proteome and Physiology. The Effect on Lifespan Depends upon Age and Sex"

_ijms, 2021, doi:10.3390/ijms222111606_

Round 1

Reviewer 1 Report

The authors have investigated the effect of exercise on the lifespan and mitochondrial proteome with physiology using D. melanogaster.
The effect of exercise is well recognized as important subject, so that using D. melanogaster is reasonable to see the relationship with lifespan.
However, the result is not easy to understand without reasonable explanation.
Based on the experimental design, early life exercise and life time exercise is the same experiment until the 14th day. It is not acceptable the big difference is seen in male experiment. These experimental data will not be reproducible. Related to this, it is unclear the reason that authors are using "survival probability". In this experiment, survival rate or survival number is easier to understand the situation, and they don't need calculation after counting. Why is survival probability needed? 

In experimental section, which exercise regime was subjected to protein analyses is not written. In discussion section, lines 462-463 says " flies that had been exercised for one week, 1-4 days post-eclosion". This condition is close to early life exercise which resulted in the shorter lifespan. This experiment is not investing the effect of exercise but the effect of harsh exercise shortening the lifespan. This difference of point of view will give another conclusion to the cause of changing mitochondrial physiology. Is it come from exercise or high stress?

Minor points,

The abbreviations are not well defined.
line 161, what is "DM" mitochondrial fraction?
line 279, 2DE-MS
Table3, Fse is not explained.

Typo lline 281, NASH should be NADH

Figure 4 presenting two A)
Left graph should be B), and its vertical label is wrong.

Experimental section should be written more in detail.
The peptide sample for MALDI-Tof seems to be from 2D-PAGE. It should be written at 2.10 or 2.9.

The name of equipment is also important. Depend on that scope of the data can change. Is it sure that de novo sequencing is done with MALDI-Tof MS/MS? 

For the LC-MS/MS experiment, which kind of electrophoresis is done before "digestion in-gel with trypsin" ?

Author Response

We would like to  thank the reviewers for the time they have taken to make thoughtful comments and suggestions. As requested, we have responded to each of the points in this letter.  We are glad the reviewers found merit in our work and hope they will agree that now we have adjusted the areas which were hampering understanding it has reached a good standard for publication.

Reviewer 1

Major comments

  1. However, the result is not easy to understand without reasonable explanation.
    Based on the experimental design, early life exercise and life-time exercise is the same experiment until the 14th day. It is not acceptable the big difference is seen in male experiment. These experimental data will not be reproducible. Related to this, it is unclear the reason that authors are using "survival probability". In this experiment, survival rate or survival number is easier to understand the situation, and they don't need calculation after counting. Why is survival probability needed?

With respect to the first point, we agree that there are potentially interesting differences in the survival outcomes to this point in the experiment. We are intrigued by many aspects of this study that we haven’t fully delved into. However, we have controlled the experiments in all the ways one would expect and have repeated them also (see line 113 methods section). However, there is a persistent and continued difference in the survival outcomes beyond this point in the experiment as well. The log-rank test p value determines this difference is statistically significant, and we therefore believe these data to be robust and should be presented.

The presentation of ‘probability of survival’ in the lifespan assays is the correct form for data output in Kaplan-Meier survival curve analysis, and as such is presented here.

  1. In experimental section, which exercise regime was subjected to protein analyses is not written. In discussion section, lines 462-463 says " flies that had been exercised for one week, 1-4 days post-eclosion". This condition is close to early life exercise which resulted in the shorter lifespan. This experiment is not investing the effect of exercise but the effect of harsh exercise shortening the lifespan. This difference of point of view will give another conclusion to the cause of changing mitochondrial physiology. Is it come from exercise or high stress?

The exercise regime used for the flies that were then subject to mitochondrial proteomic analysis was one week of exercise (the same procedure as used in the survival assays, but only for one week as opposed to two). The similarity of this to the early life exercise regime is incidental, and to exercise the flies at any stage in their lifetime would mean the experimental procedure looks somewhat similar to one of the lifespan assay procedures.

Our intention for this experiment was to characterise changes in the mitochondrial proteome caused by exercise, and by extension their physiology, in response to a sustained period of exercise. It could be argued that if we had waited until mid-life or later then we would only be exercising the ‘survivors’ to that point. For this reason, we chose the most pragmatic approach to exercise them as soon as was reasonable.

Minor comments

  1. The abbreviations are not well defined.

line 161, what is "DM" mitochondrial fraction?

line 279, 2DE-MS

Table3, Fse is not explained.

DM was replaced with Drosophila.

2DE-MS is first referred to on lines 91 and 92 in the introduction, so the first reference to the method in the introduction has been amended to say via ‘2D-gel-electrophoresis coupled with mass spectrometry (2DE-MS)’. The technique is referred to as 2DE-MS from this point onward.

Fse, as seen in table 3, is now seen in the amended legend, in line with the recommendation to amend all figure and table legends by another reviewer. The legend now describes the different groupings as ‘F/M/g/se, female/male group/single exercised; F/M/g/sne, female/male group/single non-exercised.

  1. Typo lline 281, NASH should be NADH

NASH amended to NADH.

  1. Figure 4 presenting two A)

Left graph should be B), and its vertical label is wrong.

Figure legend was amended to refer to the two graphs correctly in describing their contents, and the vertical label for the graph containing the FCR values has also been corrected.

  1. Experimental section should be written more in detail.
    The peptide sample for MALDI-Tof seems to be from 2D-PAGE. It should be written at 2.10 or 2.9.

Sections 2.9 and 2.10 have been amended for clarity, and now read as follows:

‘2.9. 2D gel electrophoresis (2D – PAGE)

Drosophila mitochondrial fractions and whole-fly homogenates were subject to isoelectric focussing using the ZOOM IPG system and pH 3 – 10 (non-linear) ZOOM IPG strips following the manufacturers protocol (Life Technologies). Gels were stained and imaged, before excision of protein spots prior to identification. Before analysis with SameSpots software (one-way Anova). Spots identified with SameSpots were excised from the gel for proteomic analysis.

2.10. Matrix-assisted laser desorption ionization tandem time-of-flight mass spectrometry (MALDI – TOF/MS)

Samples identified with SameSpots were excised from the 2D gel and were analysed at the Centre of Excellence in Mass Spectrometry at the University of York. Proteins were reduced and alkylated, followed by digestion in-gel with trypsin. MALDI-TOF/MS was sued to analyse the samples. The generated tandem MS data were compared against the NCBI database using the MASCOT search program to identify the proteins. De novo sequence interpretation for individual peptides was inferred from peptide matches.’

  1. The name of equipment is also important. Depend on that scope of the data can change. Is it sure that de novo sequencing is done with MALDI-Tof MS/MS?

Yes, MALDI-TOF MS/MS was used to identify the proteins.

  1. For the LC-MS/MS experiment, which kind of electrophoresis is done before "digestion in-gel with trypsin" ?

SDS-PAGE was the electrophoresis technique used. To reflect this, section 2.11. of the methods has been amended to say:

‘Following separation with SDS-PAGE (NuPAGE™ 4 to 12%, Bis-Tris, 1.0 mm, Mini Protein Gel, 12-well) according to the manufacturer’s protocols, the gel-resolved mitochondrial fractions were analysed by the Centre of Excellence in Mass Spectrometry at the University of York.’

We hope that we have accurately and comprehensively responded to the reviewers concerns but would be happy to expand on these points further if this is desired.

Sincerely,

On behalf of all the authors

Lisa Chakrabarti

Reviewer 2 Report

Manuscript review

Exercising D. melanogaster modulates the mitochondrial proteome and physiology - the effect on lifespan depends upon age and sex

By Brad Ebanks, Ying Wang, Gunjan Katyal, Chloe Sargent, Thomas L Ingram, Antonia Bowman, Nicoleta Moisoi and Lisa Chakrabarti

In their manuscript "Exercising D. melanogaster modulates the mitochondrial proteome and physiology - the effect on lifespan depends upon age and sex", Brad Ebanks is questioning the impact of exercise on ageing and quality of life using the fly D. melanogaster as a model. They're interrogating effects of sex and group size, at different stages of their lifetime, and recorded their lifespan. They investigate at which point in an individual's lifetime which exercise is most effective at promoting healthy ageing and whether it has a direct effect on lifespan.

The manuscript is well organized, but some experiments presented make it difficult to read (see major points) and could be presented as a supplement. It is also difficult to understand the figures because the legends are not complete, not very informative and contain some errors (see minor points).

Major points:

1 – High-resolution respirometry (line 256). This experiment needs to be better explained so that readers can understand the meaning of the different parameters measured and what this means for the mitochondria.

2 – Protein changes after one week of exercise (line 273). This experiment provides limited information, but it remains interesting, and should be presented only as a supplemental, not in the main text.

3 – Increased expression of many proteins… (line305). The legends of these experiments (Figure 5 and 7) are incomplete and therefore make interpretation difficult. For example, it is inferred that the columns in the heatmap are likely to correspond to biological replicas, but this is not stated. Furthermore, the authors do not discuss the intra-replica variations that are observed for example for NADH dehydrogenase, (2 up-regulated versus 1 down-regulated). This also applies to figure 7.

4 – Figures 6 and 8 do not add much to Tables 4 and 5 and could also be included as an additional figure.

5 – Revise all figure and table legends.

Minor points:

Line 170: “sued” should be used.

Line 262-263, figure 4: “B)” is mentioned in the legend but not in the figure (twice the A).

Line 262-263, figure 4: Pyr+Mal is shown as "ns" in Figure 4, but given the error bars presented this seems surprising.

Author Response

                                                                                                                             We would like to  thank the reviewers for the time they have taken to make thoughtful comments and suggestions. As requested, we have responded to each of the points in this letter.  We are glad the reviewers found merit in our work and hope they will agree that now we have adjusted the areas which were hampering understanding it has reached a good standard for publication.

Reviewer 2

Major comments

  1. High-resolution respirometry (line 256). This experiment needs to be better explained so that readers can understand the meaning of the different parameters measured and what this means for the mitochondria.

Section 3.2. of the results section has been amended to clarify the HRR experimental data as follows:

‘The flux control ratio (FCR) is an internal normalization of a given respiratory rate. It takes the ratio of the measured respiratory rate and the maximal uncoupled electron transport (ET) capacity of the mitochondria. We found a significant difference in the mean FCR of exercised and non-exercised flies when succinate, a complex II substrate, was added after the addition of pyruvate and malate (0.32 and 0.13, respectively; student’s t-test p-value = 0.025). Furthermore, the mean ET capacity was significantly greater in non-exercised flies compared to the exercised flies (42.55 and 14.27; student’s t-test p-value = 0.002) (Figure 4, Table 1).’

  1. Protein changes after one week of exercise (line 273). This experiment provides limited information, but it remains interesting, and should be presented only as a supplemental, not in the main text.

The MALDI-TOF MS data have been moved to a supplemental data sheet.

  1. Increased expression of many proteins… (line305). The legends of these experiments (Figure 5 and 7) are incomplete and therefore make interpretation difficult. For example, it is inferred that the columns in the heatmap are likely to correspond to biological replicas, but this is not stated. Furthermore, the authors do not discuss the intra-replica variations that are observed for example for NADH dehydrogenase, (2 up-regulated versus 1 down-regulated). This also applies to figure 7.

This legend along with others (to respond to comment 5), have been enhanced to offer greater clarity on data contained with the figures and tables of the manuscript.

With respect to the intra-replica variations, with the given example being NADH dehydrogenase, it is to be expected that in some instances that there are differences between biological replicates. The figures are presented to show the general trend of changed patterns of protein expression, with the RAR of >1.5 used as the cut-off for this. We have been careful to not overstate the significance of these findings.

However, in light of this we have amended lines 327-329 to acknowledge this variation as follows:

‘Of the 337 commonly identified proteins, 51 had increased expression (RAR>1.5) in response to exercise, with some intra-replica variation observed (Figure 5).’

  1. Figures 6 and 8 do not add much to Tables 4 and 5 and could also be included as an additional figure.

We agree that there is some redundancy in containing figures 6 and 8 alongside the corresponding tables 4 and 5. We have therefore moved figures 6 and 8 to the supplemental data sheet, with a thumbnail image of the PPI networks incorporated into the corresponding tables, 2 and 3, we think these provide an instant visualisation of the change in the protein landscape.

  1. Revise all figure and table legends.

Multiple figure and table legends amended, see ‘track changes’ in revised manuscript for full changes.

Minor comments

  1. Line 170: “sued” should be used.

Amended to read as ‘used’.

  1. Line 262-263, figure 4: “B)” is mentioned in the legend but not in the figure (twice the A).

Figure legend and figure corrected to refer to the two graphs as A and B in the correct order.

  1. Line 262-263, figure 4: Pyr+Mal is shown as "ns" in Figure 4, but given the error bars presented this seems surprising.

While there appears to be little overlap between the error bars for the exercised and non-exercised flies for this respiratory state, table 1 shows that the p-value is 0.143 (unpaired students t-test) and the difference therefore not significant at the level which is normally accepted.

We hope that we have accurately and comprehensively responded to the reviewers concerns but would be happy to expand on these points further if this is desired.

Sincerely,

On behalf of all the authors

Lisa Chakrabarti

Round 2

Reviewer 1 Report

Major comments

The same experiments should give the same result. When there is unexpectable differences, there is some uncontrolled or unconscious differences in experimental conditions. Although early life exercise and life-time exercise is the same experiment until the 14th day, their differences are quite big. Since this happened at the early days, latter data cannot be compared properly.

With the same data, this manuscript is not acceptable.

About the probability of survival, with entire data of all animals give the same result with survival rate. Kaplan-Meier survival curve analysis is needed for the estimation for censored data, so that not needed here. The result is the same but not recommendable.

Reviewer 2 Report

The changes made improve the manuscript and its comprehension. It can now be published.